# Food Preservatives Induce *Proteobacteria* Dysbiosis in Human-Microbiota Associated *Nod2*-Deficient Mice

**DOI:** 10.3390/microorganisms7100383

**Published:** 2019-09-23

**Authors:** Lucia Hrncirova, Vladimira Machova, Eva Trckova, Jan Krejsek, Tomas Hrncir

**Affiliations:** 1Charles University in Prague, The Faculty of Medicine in Hradec Kralove, 500 03 Hradec Kralove, Czech Republic; 2The Institute of Microbiology, The Czech Academy of Sciences, 549 22 Novy Hradek, Czech Republic

**Keywords:** gut microbiota, gnotobiotic, human microbiota-associated, dysbiosis, Clostridiales, Proteobacteria, diversity, antimicrobial food additives, Crohn’s disease

## Abstract

The worldwide incidence of many immune-mediated and metabolic diseases, initially affecting only the wealthy Western countries, is increasing rapidly. Many of these diseases are associated with the compositional and functional alterations of gut microbiota, i.e., dysbiosis. The most consistent markers of the dysbiosis are a decrease in microbiota diversity and an expansion of *Proteobacteria*. The role of food preservatives as potential triggers of gut microbiota dysbiosis has been long overlooked. Using a human microbiota-associated mouse model, we demonstrate that a mixture of common antimicrobial food additives induces dysbiosis characterised by an overgrowth of *Proteobacteria* phylum and a decrease in the *Clostridiales* order. Remarkably, human gut microbiota in a *Nod2*-deficient genetic background is even more susceptible to the induction of *Proteobacteria* dysbiosis by additives than the microbiota in a wild-type background. To conclude, our data demonstrate that antimicrobial food additives trigger gut microbiota dysbiosis in both wild-type and *Nod2*-deficient backgrounds and at the exposure levels reached in European populations. Whether this additive-modified gut microbiota plays a significant role in the pathogenesis of immune-mediated and metabolic diseases remains to be elucidated.

## 1. Introduction

### 1.1. Dysbiosis of Gut Microbiota

Many inflammatory, autoimmune, metabolic, neoplastic, and neurodegenerative diseases are associated with compositional and functional alterations of the gut microbiota, known as dysbiosis. Dysbiosis is usually characterised by these two major features: (1) The reduction or complete loss of commensals. The depletion of commensals leads to decreased microbiota diversity, which is associated with many immune-mediated and metabolic disorders [1]. For example, a decrease in butyrate-producing *Clostridiales* is strongly correlated with the severity of inflammatory bowel disease (IBD) [2,3]. (2) The overgrowth of potentially pathogenic microbiota (pathobionts). In a healthy gut ecosystem, pathobionts represent a relatively low percentage of gut microbiota. However, in many immune-mediated diseases, the pathobionts outgrow other commensals. For example, the outgrowth of Gram-negative bacteria from the *Enterobacteriaceae* family, a subgroup of *Proteobacteria* phylum, is frequently observed in both IBD patients [4] and mouse models of IBD [5]. The bloom of *Proteobacteria* is associated with many other diseases, and an increased abundance of *Proteobacteria* is suggested as a potential diagnostic marker of dysbiosis and the risk of disease [6].

Dysbiosis might be caused by host-derived factors such as genetic background, health status (infection, inflammation), and lifestyle habits or, even more importantly, by environmental factors such as diet (high in sugars, low in fibre), xenobiotics (antibiotics, medication, food additives, chlorinated water), or hygienic environment.

The dysbiotic microbiota can influence the host immune system and mucosal integrity via various mechanisms. These mechanisms include the modulation of inflammasome signalling through microbial metabolites, the modulation of Toll-like receptor (TLR) and NOD-like receptor (NLR) signalling, the degradation of secretory IgA (sIgA), shifting the balance between regulatory and pro-inflammatory T cell subsets, direct mucolytic activity, and others [7].

Whether dysbiosis is a direct cause of immune-mediated and metabolic diseases or merely reflects disease-associated changes in the host’s immune system remains unclear, but there are many examples including type 1 diabetes [8] and Parkinson’s disease [9] where the alterations in gut microbiota precede the onset of illness. Moreover, microbiota transfers from gnotobiotic to germ-free mice show that it is possible to transfer disease phenotype suggesting a causative role of gut microbiota in the pathogenesis of immune-mediated and metabolic disorders [10,11].

### 1.2. Food Additives and Gut Microbiota

Food additives are naturally occurring or synthetically produced substances which are added to food to prevent microbial growth or undesirable chemical changes. Additives can be used alone or in combination with other methods of food preservation. Food additives can be divided according to their function into several groups. For example, preservatives prevent or inhibit the spoilage of food due to microorganisms, sweeteners are added to foods as a sugar substitute, and emulsifiers allow water and oil to remain mixed together. Common preservatives, i.e., antimicrobial food additives, include benzoic acid and its salts, sorbic acid and its salts, sodium nitrite, calcium propionate, sulphites, and disodium ethylenediaminetetraacetate (EDTA).

Recently, several groups, including ours, published research demonstrating that food additives can alter gut microbiota. Chassaing et al. [12] found that dietary emulsifiers directly alter human gut microbiota composition. The emulsifier-modified microbiota can promote intestinal inflammation when transplanted to germ-free mice. The artificial sweetener Splenda promotes *Proteobacteria* dysbiosis and increases myeloperoxidase reactivity in ileitis-prone SAMP mice [13]. Irwin et al. [14] showed on four probiotic strains that sulphites inhibit in vitro bacterial growth at concentrations regarded as safe. Another group showed that antimicrobial biopolymer ε-polylysine could trigger transient compositional alterations in the mouse gut microbiome [15]. Lastly, we have found that some human gut microbes, including *Bacteroides coprocola*, *Clostridium tyrobutyricum*, and *Lactobacillus paracasei,* are highly susceptible to antimicrobial food additives. Interestingly, some additive combinations such as a benzoate-nitrite-sorbate mixture exert a notable synergistic effect [16]. However, in vivo studies examining the susceptibility of human gut microbiota to antimicrobial food additives are not available. 

Therefore, the main aims of this project were to test the hypothesis that antimicrobial food additives can induce a compositional alteration, i.e., dysbiosis, of human gut microbiota and also to find out whether host genotype plays any role in microbiota susceptibility to additives. To accomplish these goals, we have colonised wild-type and Nod2-deficient C57BL/6 mice with human gut microbiota, supplied them with a mixture of three commonly used antimicrobial additives, and performed detailed metagenomic analyses.

## 2. Materials and Methods

### 2.1. Animals and Experimental Design

The wild-type (stock number: 000664) and Nod2-deficient (stock number: 005763) C57BL/6 mice were acquired from a commercial vendor (Jackson Laboratories, USA) and bred for several generations in our gnotobiotic facility.

To determine the effect of antimicrobial food additives on the composition of human gut microbiota, “humanised” gnotobiotic mice, i.e., originally germ-free mice colonised with human gut microbiota, were supplied from birth with either sterile water or water supplemented with a mixture of additives. The experimental mice were supplied ad libitum with the standard breeding diet (M-Z extrudate V1126, ssniff, Germany). Second generation (F2) mice at the age of two months were used in experiments. The size of each experimental group was as follows: water-supplied wild-type (*n* = 5), additive-supplied wild-type (*n* = 6), water-supplied Nod2-/- (*n* = 8), and additive-supplied Nod2-/- (*n* = 6). The study was reviewed and approved by the Animal Care and Use Committee of the Institute of Microbiology of the CAS (approval ID: 77/2014, approval date: 2014-12-19).

### 2.2. Rederivation of Germ-Free Mice, Mouse Diet, and Sterility Checks

Wild-type and Nod2-/- C57BL/6 mice were transferred to germ-free isolators using Caesarean section and bred for several generations. The breeding colonies of germ-free mice were supplied ad libitum with sterile autoclaved water and irradiated feed pellets (M-Z extrudate V1126, ssniff, Germany). The sterility of isolators (faecal samples, bedding, cotton swabs) was tested weekly (both aerobic and anaerobic cultures).

### 2.3. Colonisation of Germ-Free Mice with Human Gut Microbiota and Experimental Diet

Adult germ-free mice were colonised with the faecal gut microbiota of a healthy middle-aged Caucasian donor with no history of autoimmune diseases. The donor participating in the study signed and dated Informed Consent and Faecal Donor Agreement forms. The faecal sample was diluted in reduced phosphate-buffered saline (PBS) in an anaerobic chamber and introduced by gavage into germ-free animals. The procedure was completed within one hour of the collection. The human microbiota-associated mice were bred in plastic Trexler-type isolators and supplied water ad libitum. To assure the proper establishment of the microbiota along the GI tract, animals from later generations were used. Turnbaugh et al. [17] showed that the human gut microbiota could be successfully transferred from generation to generation without a significant drop in diversity. All mice were supplied ad libitum with the standard breeding diet (M-Z extrudate V1126, ssniff, Germany).

### 2.4. Antimicrobial Food Additives and Their Supplementation

To test the effect of antimicrobial food additives on the composition of gut microbiota, we decided to select the three most frequently used additives, namely sodium benzoate (E211; Cat. #71300), sodium nitrite (E250; Cat. #237213), and potassium sorbate (E202; Cat. #85520). All additives were supplied by Sigma-Aldrich, USA. In preliminary testing, we found that both autoclaving and sterile filtration decreases the efficacy of additives, so in all experiments, untreated additives were used. All additives were sterility tested (no growth in VL broth) upon arrival. The exposure to additives, normalised to mouse weight and water intake, was adjusted to match the maximum estimated daily intake of additives in European populations (Source: Report from the Commission on Dietary Food Additive Intake in the European Union, https://publications.europa.eu). Specifically, 4.8 mg/kg bw/day for benzoate, 0.36 for nitrite, and 19.0 for sorbate. The additive solutions were freshly prepared each week and supplied in drinking water ad libitum.

### 2.5. Genomic DNA Extraction

Fresh faecal samples of 2-month old HMA wild-type and Nod2-/- experimental mice were collected into 2.0-mL pre-cooled eppendorf tubes and kept at −80 °C till analysis. The bacterial genomic DNA was extracted using the PowerFecal DNA isolation kit (Cat. #12830, MO BIO Laboratories, USA). The humic acids which might interact with PCR amplification were removed using PowerClean DNA Clean-Up kit (Cat. #12877, MO BIO, USA). The DNA quality was checked using a standard agarose gel electrophoresis. All samples were diluted to a concentration of 30 ng/µL in the MO BIO’s elution buffer C6 (10 mM Tris) and stored at −80 °C until use in PCR.

### 2.6. PCR and Sequencing

PCR of the V3-V4 regions of the 16S rRNA gene and sequencing were performed on the Illumina MiSeq platform following the original Earth Microbiome Project protocols (http://www.earthmicrobiome.org/protocols-and-standards/) initially described by Caporaso et al. [18]. Sequencing was performed using paired-end 150 base reads.

### 2.7. Metagenomic Analysis

Microbiome bioinformatics was performed using QIIME2 2019.4 [19]. Briefly, raw sequence data were demultiplexed and quality filtered using the q2-demux plugin followed by denoising with DADA2 [20] (via q2-dada2) to identify all observed amplicon sequence variants (ASV). All ASVs were aligned with MAFFT [21] and used to construct a phylogenetic tree with FastTree 2 [22]. Alpha-diversity metrics (observed ASVs, Faith’s Phylogenetic Diversity [23], and Jaccard distance) and principal coordinate analysis (PCoA) were estimated using q2-diversity after samples were rarefied to 22,000 sequences per sample. The taxonomy was assigned to ASVs using the q2-feature-classifier [24] against the Greengenes 13_8 99% OTUs reference sequences [25].

Gneiss analysis was used to identify differentially abundant taxa among groups. To facilitate the analysis, the taxa with a per-sample frequency lower than ten were filtered out. Principal balances were obtained via Ward’s hierarchical clustering using the correlation-clustering command. The log ratios between groups at each node of the tree were calculated using the ilr-hierarchical command. A multivariate response regression model was created by running a linear regression separately on each balance using the ols-regression command. The contributions of genotype, treatment, and gender as covariates to the overall community variation were visualised through a regression summary and dendrogram heatmaps. Balances significantly affected by the covariates were identified as those with a *p*-value less than 0.05.

### 2.8. Statistics

The significance of differences in alpha diversity was assessed using a Kruskal–Wallis test with a false discovery rate (FDR) correction, and the significance of differences in beta diversity was assessed using a PERMANOVA analysis [26]. Significant balances in Gneiss analysis were determined by ANOVA. A *p*-value of ≤0.05 was regarded as significant. The graphs were generated using QIIME2 View web interface or Prism GraphPad 8.0 and annotated in Affinity Designer software.

## 3. Results

### 3.1. Body Weight, Water and Food Consumption

Neither additive administration nor genotype had any statistically significant effect on mouse weight, water intake, or pellet consumption (Figure 1).

### 3.2. Alpha Diversity

The effects of additives on alpha diversity, i.e., within-sample diversity, were determined using three alpha diversity metrics—observed ASVs, Faith’s phylogenetic index (richness), and Pielou’s index (evenness). Observed ASVs is a qualitative measure of community richness, Faith’s phylogenetic index is a qualitative measure of community richness that incorporates phylogenetic relationships between the ASVs, and Pielou’s index is a measure of community evenness. The richness is the number of different species in a sample, and the evenness is how balanced they are to each other.

Additive treatment decreased the number of ASVs in the gut microbiota of both wild-type and Nod2-/- mice to an average of 70 ASVs. The original number of ASVs was slightly higher in wild-type mice compared to Nod2-/- mice, precisely 94 ASVs and 88 ASVs. The decrease in alpha diversity as quantified by Faith’s phylogenetic index was significant only for the microbiota of wild-type mice (*H* = 4.8, *p* = 0.028, Figure 2A). There was no significant change in microbiota evenness (Pielou’s index) in either wild-type or Nod2-/- mice (Figure 2B).

### 3.3. Beta Diversity

The effects of additives on beta diversity, i.e., between sample diversity, were determined using Jaccard distance metrics, which is a qualitative measure of community dissimilarity.

The principal coordinate analysis showed that the genotype, i.e., Nod2-deficiency, significantly influences gut microbiota composition. The microbiota of wild-type and Nod2-deficient mice clustered separately (*p* = 0.001 for unweighted UniFrac distance). This finding is not novel. Previously, it had been demonstrated, in cooperation with our group, that Nod2 is essential in the regulation of commensal microbiota [27].

However, both wild-type and Nod2-/- microbiota were highly susceptible to additive treatment. Interestingly, the microbiota of Nod2-/- mice (*p* = 0.001) was more susceptible to the disturbance by additives compared to the microbiota of wild-type mice (*p* = 0.005). The remarkable separation of control and additive-treated microbiota is visible in Figure 3.

### 3.4. Taxonomic Analysis/Relative Abundance

Taxonomic analysis at the phylum level (Figure 4) shows that the microbiota of Nod2-/- mice is more susceptible to additive treatment compared to the microbiota of wild-type mice. The most representative additive-induced changes in relative abundance in both wild-type and Nod2-/- mice were a decrease in *Firmicutes* (wild-type, 31.0 to 21.4%; Nod2-/-, 45.0 to 28.1%) and an increase in *Proteobacteria* (wild-type, 4.0 to 5.4%; Nod2-/-, 2.1 to 9.0%). The additive treatment also caused the expansion of *Verrucomicrobia* (20.1 to 29.8%) in wild-type mice and *Bacteroidetes* in Nod2-/- mice (49.4 to 60.2%). In control mice (water), the effect of Nod2-deficiency was demonstrated mostly by the expansion of *Bacteroidetes* and *Firmicutes* at the expense of *Verrucomicrobia*. 

At the order level (Figure 5), the most representative effect of additive treatment was a decrease in the relative abundance of *Clostridiales* (*Firmicutes* phylum; wild-type, 30.7 to 20.9%; Nod2-/-, 44.9 to 26.4%) and an increase in *Burkholderiales* (*Proteobacteria* phylum; wild-type, 1.1 to 3.0%; Nod2-/-, 1.1 to 6.5%) in both wild-type and Nod2-/- mice. The drop in *Clostridiales* in wild-type mice was mostly compensated for by the expansion of *Verrucomicrobiales* (20.1 to 29.8%). Interestingly, in Nod2-deficient mice, the decrease of *Clostridiales* was compensated for by the expansion of *Bacteroidales* (49.4 to 60.2%) and *Burkholderiales* (1.1 to 6.5%).

### 3.5. Differential Abundance Analysis (Gneiss Analysis)

The differential abundance analysis using the Gneiss method confirmed the significant effect of additive treatment on gut microbiota in both wild-type (Figure 6) and Nod2-/- mice (Figure 7). The overall linear regression model fit was R^2^ = 0.44 with the covariate “treatment” accounting for 32.3% of the community variation. In both wild-type and Nod2-/- mice, the log ratios of balances y0 (wild-type, β = −20.6, *p* < 0.001; Nod2-/-, β = 33.8, *p* < 0.001) and y2 (wild-type, β = 13.9, *p* < 0.001; Nod2-/-, β = −22.4, *p* < 0.001) were significantly different between control and additive-treated samples. 

For wild-type mice, the log ratio of balance y0 was higher in additive-treated samples compared to control samples showing that the relative abundance of y0 denominator ASVs was lower in the additive-treated microbiota compared to the control water-supplied microbiota (Figure 8A). The decrease in the log abundance of these ASVs is well seen in the dendrogram heatmap (Figure 6). The ASVs identified as depleted in the additive-treated microbiota were assigned mostly to *Lachnospiraceae* (*n* = 13), *Ruminococcaceae* (*n* = 7), and *Bacteroidaceae* (*n* = 6). Other ASVs were assigned to *Veillonelaceae* (*n* = 1), *Clostridiaceae* (*n* = 1), *Peptococcaceae* (*n* = 1), and *Alcaligenaceae* (*n* = 1) (Table 1 and Appendix A). Conversely, the log ratio of balance y2 was lower in additive samples compared to control samples indicating that the relative abundance of y2 denominator ASVs was higher in the additive-treated microbiota (Figure 8C). The increase in the log abundance of these ASVs is visible in the dendrogram heatmap (Figure 6). The ASVs identified as more abundant were assigned to *Bacteroidaceae* (*n* = 4), *Lachnospiraceae* (*n* = 2), *Turicibacteraceae* (*n* = 1), *Ruminococcaceae* (*n* = 1), and *Alcaligenaceae* (*n* = 1) (Table 1 and Appendix A).

For Nod2-/- mice, the log ratio of balance y0 was higher in control samples compared to additive-treated samples indicating that the relative abundance of y0 denominator ASVs was higher in the additive-treated microbiota than control water-supplied microbiota (Figure 8B). This increase in the log abundance of the y0 denominator ASVs is visible in the dendrogram heatmap (Figure 7). The expanded ASVs from the y0 denominator were mostly assigned to *Lachnospiraceae* (*n* = 12), *Bacteroidaceae* (*n* = 7), and *Ruminococcaceae* (*n* = 4). Other ASVs were assigned to *Desulfovibrionaceae* (*n* = 2), *Rikenellaceae* (*n* = 2), *Erysipelotrichaceae* (*n* = 2), *Clostridiales* (*n* = 2), *Porphyromonadaceae* (*n* = 1), *Enterobacteriaceae* (*n* = 1), and *Turicibacteraceae* (*n* = 1) (Table 1 and Appendix A). Conversely, the log ratio of balance y2 was lower in control samples compared to additive-treated samples indicating that the relative abundance of y2 denominator ASVs was lower in the additive-treated samples compared to control samples (Figure 8D). This decrease in the log abundance of y2 denominator ASVs is visible in the dendrogram heatmap (Figure 7). These depleted ASVs were mostly assigned to *Lachnospiraceae* (*n* = 20), *Ruminococcaceae* (*n* = 8), and *Clostridiales* (*n* = 7). The remaining ASVs were assigned to *Bacteroidaceae* (*n* = 2) and *Rikenellaceae* (*n* = 2) (Table 1 and Appendix A).

## 4. Discussion

The main aim of the present study was to determine whether antimicrobial food additives administered at the exposure levels reached in EU populations can trigger the dysbiosis of human gut microbiota and also whether the host genotype specifically Nod2-deficiency has any effect on the gut microbiota’s susceptibility to additives.

To accomplish this goal, we have colonised germ-free C57BL/6 mice with gut microbiota isolated from a healthy human donor. These human microbiota-associated (HMA) experimental mice of wild-type and Nod2-/- genotype, an animal model for Crohn’s disease, were provided with either sterile water or water supplemented with a mixture of the most commonly used antimicrobial food additives (AMFA) including sodium benzoate, sodium nitrite, and potassium sorbate. The HMA mice of both genotypes were randomly allocated to control, i.e., water supplied, and additive-treated groups. The size of each experimental group was 5 to 8 mice. The metagenomic analysis of faecal samples was performed using a QIIME2 software package.

Bioinformatic analysis revealed that the mixture of additives decreased the number of ASVs initially present in both wild-type and Nod2-/-mice. The decrease in alpha diversity of faecal samples as represented by Faith’s index was significant only for wild-type mice (Figure 2A). However, it is essential to note that the number of ASVs in control Nod2-/- mice (*n* = 88) was lower when compared to control wild-type mice (*n* = 94), and the additive treatment decreased the average number of ASVs in both genotypes to 70 ASVs. Therefore, the effect of additive treatment on alpha diversity in Nod2-/- mice was partly limited by an already decreased number of ASVs due to the genotype effect.

Analysis of beta diversity confirmed a significant effect of genotype on the composition of gut microbiota. The finding that Nod2 is a major factor in the regulation of commensal microbiota is not novel and was published in cooperation with our group a long time ago [27]. However, this project focused primarily on the possible dysbiosis-inducing capacity of additives in two genetic backgrounds, i.e., wild-type and Nod2-deficient. In this regard, the beta diversity analysis demonstrates that human gut microbiota in both wild-type and Nod2-/- settings is susceptible to additives. Remarkably, the gut microbiota in Nod2-/- environment (*p* = 0.001) was significantly more susceptible to additives compared to the microbiota in a wild-type environment (*p* = 0.005). This finding could be explained by impaired defensin production in Nod2-/- mice which renders their microbiota more susceptible to environmental triggers. The tight clustering of samples in all four experimental groups and the separation of control and additive-treated samples are convincingly documented by the PCoA plot (Figure 3). 

The taxonomic analysis of the samples from wild-type mice showed that the effect of additives was mostly represented by a decrease in the relative abundance of *Firmicutes* and an increase in *Verrucomicrobia* and *Proteobacteria.* The abundance of *Bacteroidetes* was almost unaffected. The analysis of the additive-treated samples from Nod2-/- mice also showed a decrease in *Firmicutes* and an increase in *Bacteroidetes* and *Proteobacteria*. Therefore, the common feature in both wild-type and Nod2-/- samples is a depletion of *Firmicutes*, mostly from the *Clostridiales* order, and the expansion of *Proteobacteria*, mostly from the *Burkholderiales* order.

The differential abundance analysis gives a more dynamic image of the effect of additives on gut microbiota than simple taxonomic analysis. For example, in wild-type mice, the seemingly unharmed *Bacteroidales* are actually affected (Figure 6). Six ASVs assigned to the *Bacteroidaceae* family are depleted, and simultaneously four ASVs assigned to the same family are expanded, resulting in an unaltered relative abundance at the phylum or order level (Table 1). Moreover, the total depletion of the *Clostridiales* order is compounded of a decrease of 23 ASVs and an increase of four ASVs (Figure 6 and Table 1). The same is true for Nod2-/- mice, where the expansion of *Bacteroidetes* consists of an increase of 10 ASVs assigned to the *Bacteroidaceae* family, and at the same time, a decrease of 4 ASVs assigned to the same family, and the depletion of *Clostridiales* consists of an increase of 16 ASVs and a decrease of 35 ASVs (Figure 5 and Table 1). Notably, the expansion of *Proteobacteria* in Nod2-/- mice consisted solely of an increase of three ASVs with no *Proteobacteria* depletion.

Therefore, it is important to note that not all ASVs in particular taxonomic levels and categories are susceptible or resistant to the effect of additives, and the overall abundance as represented by the taxonomic bar plot (Figure 5) consists of depleted and expanded ASVs.

Published research on the effects of antimicrobial food additives on human gut microbiota, i.e., preservatives, is scarce. Irwin et al. [14] showed on four probiotic strains that sulphites may in vitro inhibit bacterial growth at concentrations regarded as safe. Another group showed that antimicrobial biopolymer ε-polylysine could trigger transient compositional alterations in the mouse gut microbiome [15]. Our recent in vitro research on human gut isolates demonstrates that some gut microbes are highly susceptible to commonly used antimicrobial food additives. For example, the growth of *Bacteroides coprocola* is inhibited by 50% in the presence of only 30 ng ml-1 of sodium nitrite. Other highly susceptible strains include *Clostridium tyrobutyricum* and *Lactobacillus paracasei.* Alarmingly, some combinations of additives, such as a benzoate-nitrite-sorbate mixture, showed a very high degree of synergism [16]. However, the research on non-antimicrobial additives presented some notable findings recently. Chassaing et al. [12] showed that food emulsifiers alter human gut microbiota composition and may promote colitis and metabolic syndrome, and Rodriguez-Palacios et al. [13] reported that the artificial sweetener Splenda can induce *Proteobacteria* dysbiosis.

The limitations of the work are as follows: firstly, the gut microbiota isolated from other donors, i.e., dissimilar as to gender, age, or race, might exhibit different levels of susceptibility to additives. Ideally, the metagenomic analysis should be performed on a higher number of donor samples, but the upscaling of this work was not possible due to a limited project budget. Secondly, the gut microbiota acquired from a human donor and transplanted into germ-free mice does not represent an identical copy of the original microbiota due to several factors including the transient exposure to oxygen during transfer or different luminal conditions, such as pH, bile acids, IgA specificity, and defensins. Finally, the gut microbiota isolated from a European donor already exhibits a reduced diversity—the most sensitive strains have already been lost—due to the exposure to antibiotics, medications, and food additives. Therefore, we speculate that the gut microbiota isolated from donors living in less-developed rural areas may be even more susceptible to additive exposure.

Moreover, it is important to note that the observed effects of additives on human gut microbiota might be caused by a single additive as well as any combination of additives. Therefore, further experiments testing each additive separately are needed. Most importantly, further experiments should focus on the extensive analysis of direct or indirect, via microbiota, effects of additives on host physiology including gut permeability, immune system parameters, histology, serum parameters, and the bioavailability of additives. 

To sum up the findings of this project, we conclude that the mixture of commonly used antimicrobial food additives at the exposure levels reached in EU populations has the potential to induce human gut microbiota dysbiosis characterised by a decrease in microbiota diversity, a depletion of the *Clostridiales* order, and the expansion of *Proteobacteria* phylum and, that the Nod2-/- genotype is particularly susceptible to gut microbiota disruption. These findings are highly important because this type of dysbiosis is associated with many immune-mediated and metabolic disorders.

## Figures and Tables

**Figure 1 microorganisms-07-00383-f001:**
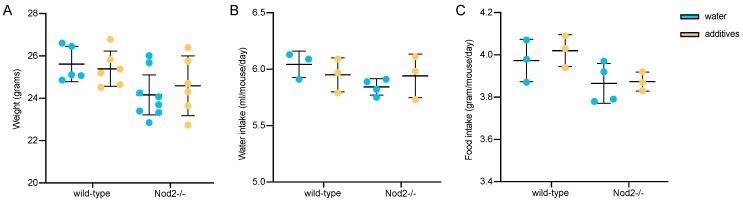
The scatter dot plots show the effect of the additive treatment on (**A**) individual mouse weight at the end of experiment, (**B**) average water intake per mouse and day, and (**C**) average feed pellet consumption per mouse and day. The water intake and pellet consumption data were recorded during the last week of experiment. Each data point in (**B**) and (**C**) shows average per mouse consumption in every animal cage. The horizontal lines represent mean plus/minus one standard deviation. The statistical significance was determined using one-way ANOVA followed by Tukey’s multiple comparisons test.

**Figure 2 microorganisms-07-00383-f002:**
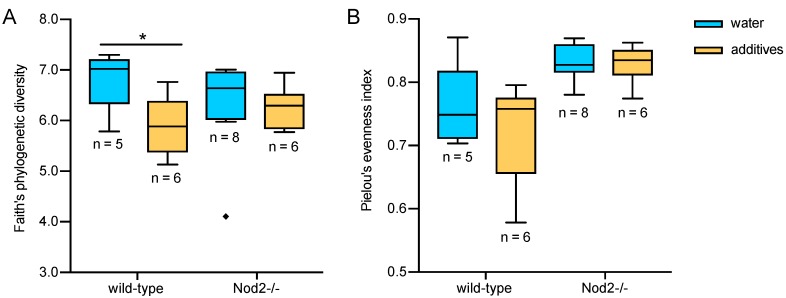
The alpha diversity box-plots show the effect of the additive treatment on the (**A**) richness (Faith’s phylogenetic diversity) and (**B**) evenness (Pielou’s index) of the gut microbiota in wild-type and Nod2-/- mice. The additives significantly decreased the richness of the microbiota in wild-type mice (*p* = 0.0285). The richness of the microbiota of Nod2-/- mice and evenness were not significantly influenced. Boxes represent the interquartile range (IQR), i.e., the difference between the 25th and 75th percentiles, and the horizontal line inside the box is the median. Whiskers stop at the lowest and highest values within 1.5 times the IQR from the first and third quartiles, respectively. “◆” indicates an outlier, i.e., a value lower than the 25th percentile minus 1.5 IQR. Statistical significance was determined using the Kruskal–Wallis test; **p* < 0.05.

**Figure 3 microorganisms-07-00383-f003:**
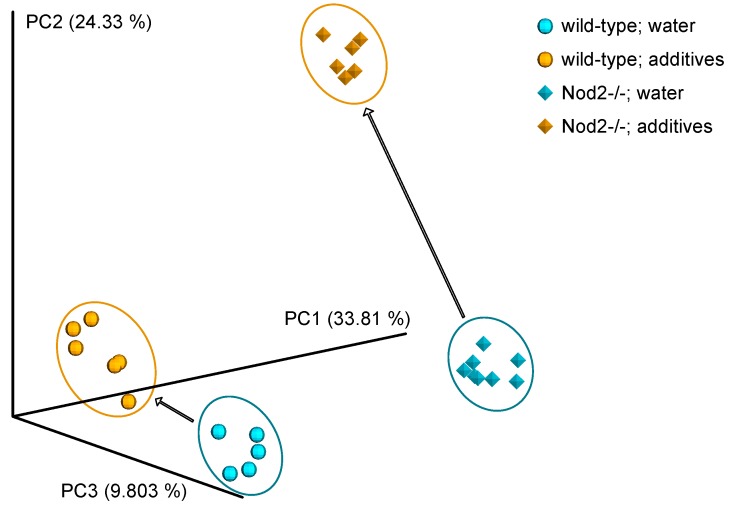
Principal coordinate analysis (PCoA) plot shows the difference in beta diversity (Jaccard) between the microbiota of wild-type and Nod2-/- mice and also the effect of additives on the microbiota composition. Each point represents the microbiota of single mouse, and the distance between points represents the dissimilarity of the microbiota. The axis percentages indicate the proportion of the data variation captured by each principal coordinate. The most contributing variable to PC1 is genotype and to PC2 and PC3 treatment.

**Figure 4 microorganisms-07-00383-f004:**
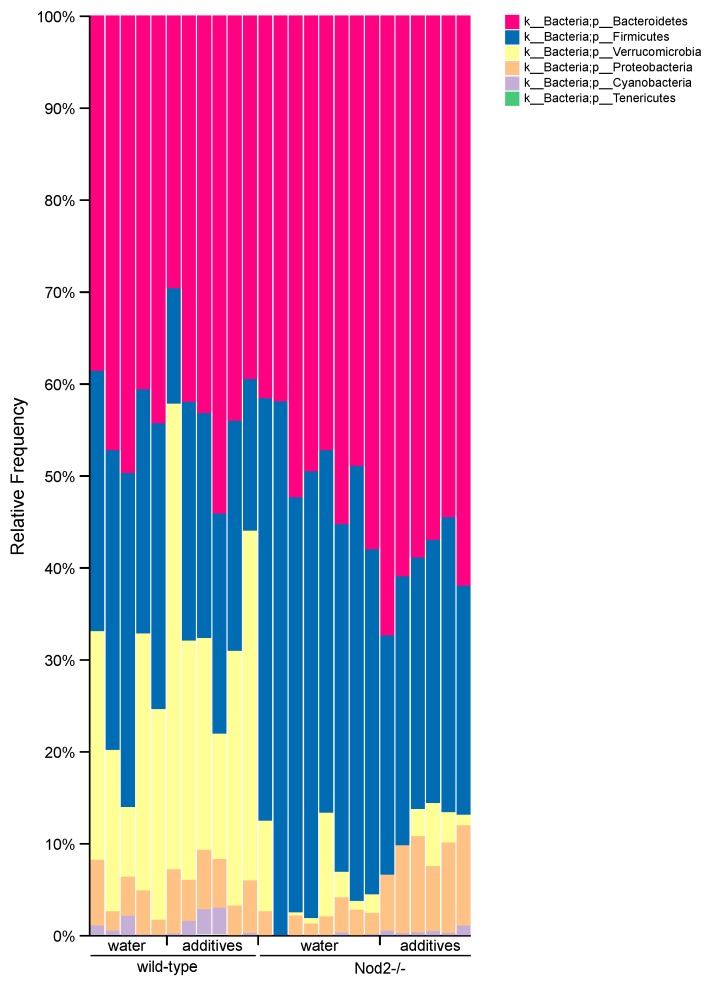
Taxonomic bar plots show the relative abundance of bacteria at the phylum taxonomic level. The data are sorted by genotype (wild-type or Nod2-/-) and treatment (water or additives).

**Figure 5 microorganisms-07-00383-f005:**
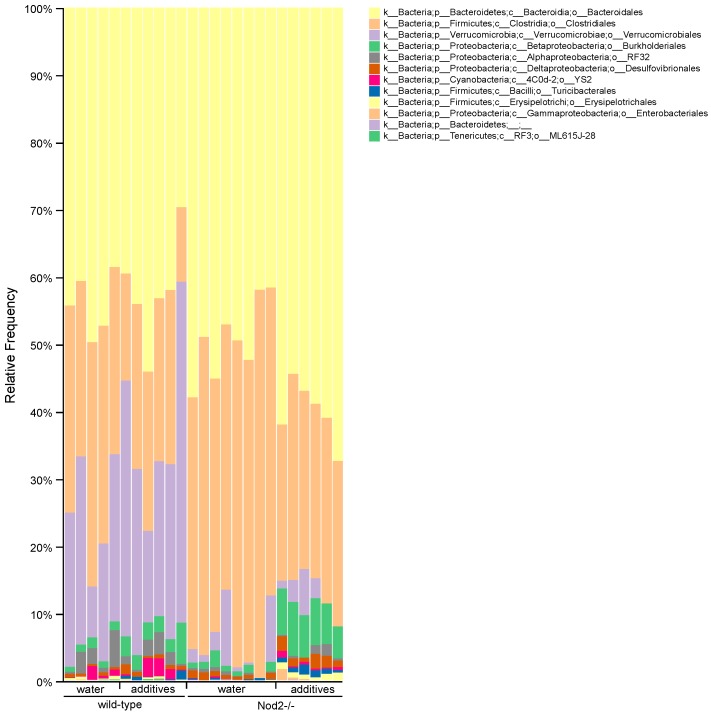
Taxonomic bar plots show the relative abundance of bacteria at the order taxonomic level. The data are sorted by genotype (wild-type or Nod2-/-) and treatment (water or additives).

**Figure 6 microorganisms-07-00383-f006:**
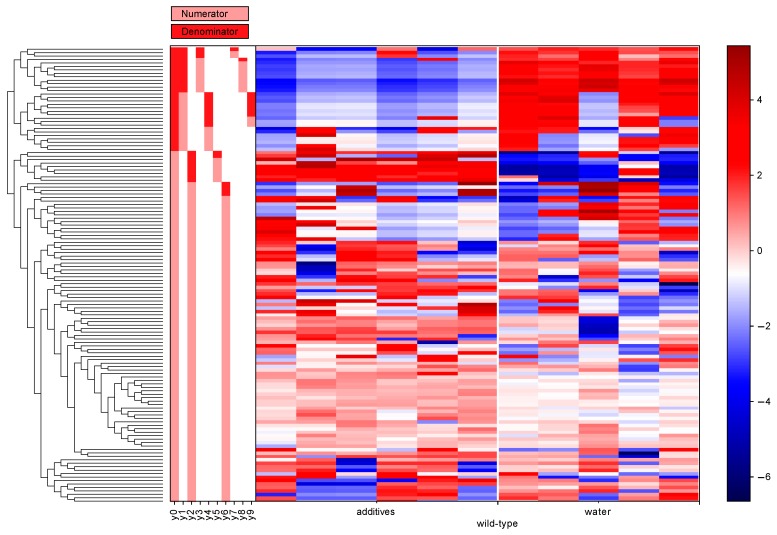
A dendrogram heatmap shows the effect of additives on the log abundance of ASVs in the gut microbiota of wild-type mice. Differences in relative abundance between control (water) and additive-treated microbiota are visible in the balance y0 and y2 (Gneiss analysis).

**Figure 7 microorganisms-07-00383-f007:**
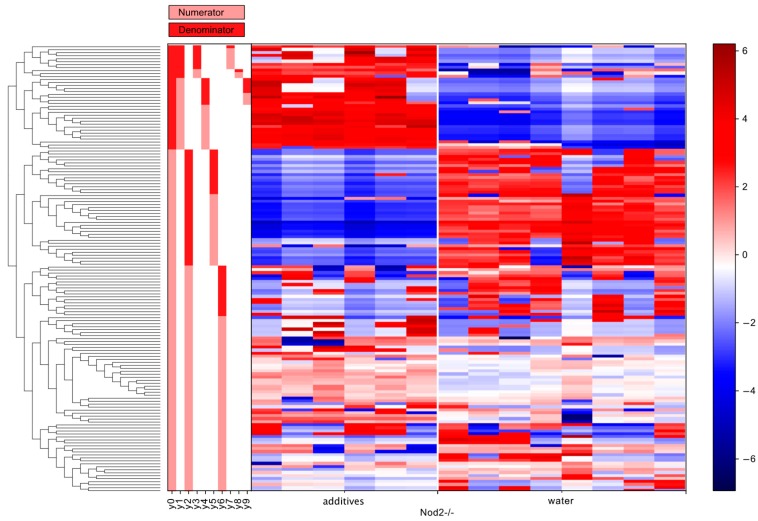
A dendrogram heatmap shows the effect of additives on the log abundance of ASVs in the gut microbiota of Nod2-/- mice. Differences in relative abundance between control (water) and additive-treated microbiota are visible in the balance y0 and y2 (Gneiss analysis).

**Figure 8 microorganisms-07-00383-f008:**
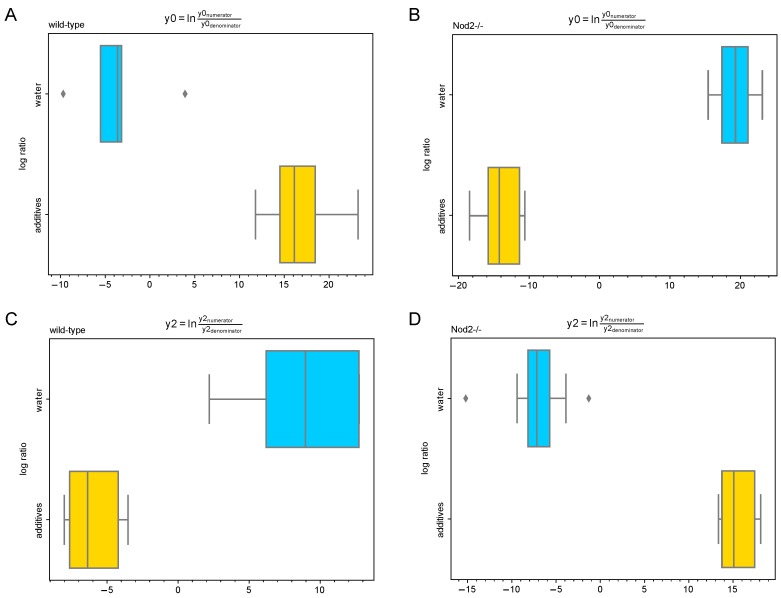
Log-ratios of balances significantly different among control and additive-treated groups. Balances (**A**,**B**) y0 and (**C**,**D**) y2 in both wild-type and Nod2-/- groups were identified by Gneiss analysis as significantly different. A lower log-ratio shows a shift in the balance toward denominator ASVs, while a higher log-ratio shows a shift toward numerator ASVs.

**Table 1 microorganisms-07-00383-t001:** Taxonomy of ASVs contributing to the balances significantly affected by the additive treatment.

Genotype	Balance of Interest	Family	Number of ASVs
*wild-type*	*y0 denominator*	*Lachnospiraceae*	13
		*Ruminococcaceae*	7
		*Bacteroidaceae*	6
		*Veillonellaceae*	1
		*Clostridiaceae*	1
		*Peptococcaceae*	1
		*Alcaligenaceae*	1
	*y2 denominator*	*Bacteroidaceae*	4
		*Lachnospiraceae*	2
		*Turicibacteraceae*	1
		*Ruminococcaceae*	1
		*Alcaligenaceae*	1
*Nod2-deficient*	*y0 denominator*	*Lachnospiraceae*	12
		*Bacteroidaceae*	7
		*Ruminococcaceae*	4
		*Desulfovibrionaceae*	2
		*Rikenellaceae*	2
		*Erysipelotrichaceae*	2
		*Clostridiales (order)*	2
		*Porphyromonadaceae*	1
		*Enterobacteriaceae*	1
		*Turicibacteraceae*	1
	*y2 denominator*	*Lachnospiraceae*	20
		*Ruminococcaceae*	8
		*Clostridiales (order)*	7
		*Bacteroidaceae*	2
		*Rikenellaceae*	2

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
