# Peer review of "Food Preservatives Induce Proteobacteria Dysbiosis in Human-Microbiota Associated Nod2-Deficient Mice"

_microorganisms, 2019, doi:10.3390/microorganisms7100383_

Round 1

Reviewer 1 Report

Authors studied the effect of three common food additives on gut microbiota. The design of the animal experiment, where fecal transplantation from a human donor was used, increase the overall quality of the study. The microbial analyses are very extensive and robust.

 Major comments:

In the Material and Method section, specifically in section 2.1 and 2.4, I did not find any information about how many animals were used per group and how long the study was conducted. I am assuming that this study is the primary study, and therefore, basic physical data (Body weight, food and water consumption) of the animals should be provided in the Result section. Are there any differences in their body weight, food and water intake? Additionally, I did not find any information about the diet given to the animals. Detail information on their treatment diet should also be included in section 2.1. The effects or dysbiosis observed in this study might be due to either one or two or a combination of all the three food additives. As the authors did not conduct independent treatment effects of each ingredient separately, it is unclear to conclude that all the three food additives were equally involved in the dysbiosis process. This limitation should be discussed. Also, please provide supportive evidence from other studies (if there is any) which indicated that all three food additives have been the potential to alter gut microbiota independently. Blood analysis to quantify all the three food additives will provide a clear picture of the absorption, bioavailability and blood level of those additives. If the authors haven’t done this, it should be indicated in the limitation as well. If the authors have removed any outliers, it should be mentioned. I see that there are only 5 animals in the wild-type water group.

Minor comments:

Line 33, ‘The overgrowth……… pathogenic Commensals. I suggest replacing the word ‘commensal’ with ‘microbiota’. Because the word commensal is more preferred when there is no harm between the host and parasite. Please remove the paragraph of line 321-326. It is not related to the current study.

Reviewer 2 Report

The article presents a very interesting analysis of the impact of additives on microbiote composition. The paper is well written and the results are of importance regarding the use of additives.

However to improve the manuscript some questions should be clarified.

The authors should precise the nature of the Nod2 mutation (i.e missense, frameshift…) the codon affected and the protein domain altered.

The authors should provide evidences that Human microbiota in humanised gnotobiotic mice remains stable over time (before food additive exposure) and similar to the original Human microbiota (i.e from the donor).

The authors should document in the material and method section that an ethical committee (IRC) gave an approval to the participation of the healthy middle-aged Caucasian donors, rather than at the end of the manuscript.

In § 3.1 the authors wrote that « Additive treatment decreased the number of ASVs in the gut microbiota of both wild-type and 
Nod2-/- mice to an average of 70 ASVs. The original number of ASVs was slightly higher in wild-type mice compared to Nod2-/- mice, precisely 94 ASVs and 88 ASVs ». Where the ASV values come from ? The reader may think that ASV are the y values of figure 1 and thus the text refers to an average of 70 ASVs in wt mice, although in figure 1 the mean (median) is at 7.0. A similar discordance may be pointed out for Nod2 -/- mice. This should be clarified.

For a better readability of the results §3.1 and of figure 1, the authors should explain what is the evenness. Likewise the number of mice in each group should be displayed on the figure 1. This would allow a direct feeling of the robustness of the data for the reader.

For figure 1B the authors wrote that the lack of significant difference for evenness between wt mice with and without additives should be a lack of power in relation with a low sample size. This explanation may not be accepted for several reasons. First the median values (in box plots) are quite similar : 0.75 (according to the slight difference between medians a very high number of mice should be requested to reach significance if a true difference exists*), and second, the number of mice is comparable with those from the ASV panel (fig 1A). The authors should better conclude to a lack of evenness difference. This would mean that although the number of species is significantly lower in additives treated mice, the number of bacteria in each phylum remains comparable (i.e. there is no phylum with an exaggerrated higher or lower number of bacteria).

*graphically assuming in wt mice, median values of 0.74 and 0.76 respectively in water and additives treated, as SD=interquartile range/1.35 : 0.1/1.35 = 0.074, a sample size calculation with these parameters shows that 215 mice per group (total 430) should be requested to have 80% power to detect such a difference (if exists) at the 0,05 level.

The authors should clearly explain for non environmentalists readers what are alpha and beta diversities.

Regarding figure 2, the authors should present the list of variables that were included in the ACP analyses, and what variables are predominant on each axe PC1, PC2, PC3.

How the authors explain that analyses at the phylum level (figure 3) would reveal an expansion of Verrucomicrobiota in wt mice and Bacteroidetes in Nod2 -/- mice, although previous results (fig 1) were in agreement with a lack of difference in evenness.

From figure 3 it seems possible that dysbiosis is present in additive treated wt mice (as compared with water treated wt mice) and is supported by Verrucomicrobiota and Bacteroidetes. However this additive induced dysbiosis is not comparable with dysbiosis of the water treated Nod2 -/- mice (i.e « Crohn like » mice) characterized with Firmicutes expansion.

Regarding figure 4 the difference for Clostridiales is not evident in wt mice : have the authors evidences (statistical ?) that 29.7% is truely different from 20.5% ? What objective reasons or arguments the authors have to conclude that this difference is not hazard related. For Nod2 -/- a visual difference appears between water and additive treated mice. However the question of objective reasons or arguments the authors have to conclude that this difference is not hazard related, may be pointed out.

In the discussion section, the authors wrote that « the mixture of additives decreased the number of ASVs 
initially present in both wild-type and Nod2-/- mice » 
however in figure 1A there is no statistical difference regarding ASV. The sentence should be amended.

Round 2

Reviewer 2 Report

Dear Author,

Regarding our question about the nature of the Nod2 mutation in your mouse strain, your answer is greatly incomplete and giving a web link implying that the reviewer has to do the job if he wants an answer to his question. You should have at least write the characteristics of the mouse modified gene and what are its consequences. On this point you have been lazy.

For future publications and answers to the reviewers I suggest you be careful not to do such clumsiness.

Anyway I spent the time to collect information on the web site you reported. But definitively I should have appreciated you did it. Finally you did not answer to my question, but I found myself the answer. And it seems that this answer is correct. Thus I have no more doubt on the validity of the mouse model in the background of your paper.

Regarding all other questions I mentioned in my first report, the answers are quite suitable. The manuscript has been edited accordingly.

Best regards

Author Response

Dear Reviewer,

We have made a sincere effort to answer reviewers' questions in the best possible way. However, we agree with the Reviewer that our response concerning the Nod2 mutation lacked in-depth and detail.

So, in the revised version of the manuscript, the generation of Nod2-deficient mice and the consequences of Nod2 receptor dysfunction are now clearly described in section 2.1.

Also, we would like to emphasize that we much appreciate the reviewers' valuable comments. Next time, we will do a better job answering ALL reviewers' comments in detail and hope to make the reviewers' work as easy as possible.

The authors sincerely apologize to the Reviewer and thank him/her deeply for the time and effort spent to review our manuscript,

With kind regards,

Authors